# Oriented electron transmission in polyoxometalate-metalloporphyrin organic framework for highly selective electroreduction of $CO_2$

Yi-Rong Wang[1], Qing Huang[1], Chun-Ting He[2], Yifa Chen[1], Jiang Liu[1], Feng-Cui Shen[1] & Ya-Qian Lan [1]

The design of highly stable, selective and efficient electrocatalysts for $CO_2$ reduction reaction is desirable while largely unmet. In this work, a series of precisely designed polyoxometalate-metalloporphyrin organic frameworks are developed. Noted that the integration of {ε-$PMo_8{}^{V}Mo_4{}^{VI}O_{40}Zn_4$} cluster and metalloporphyrin endows these polyoxometalate-metalloporphyrin organic frameworks greatly advantages in terms of electron collecting and donating, electron migration and electrocatalytic active component in the $CO_2$ reduction reaction. Thus-obtained catalysts finally present excellent performances and the mechanisms of catalysis processes are discussed and revealed by density functional theory calculations. Most importantly, Co-PMOF exhibits remarkable faradaic efficiency (>94%) over a wide potential range (−0.8 to −1.0 V). Its best faradaic efficiency can reach up to 99% (highest in reported metal-organic frameworks) and it exhibits a high turnover frequency of 1656 h$^{-1}$ and excellent catalysis stability (>36 h).

[1] Jiangsu Collaborative Innovation Centre of Biomedical Functional Materials, Jiangsu Key Laboratory of New Power Batteries, School of Chemistry and Materials Science, Nanjing Normal University, 210023 Nanjing, China. [2] MOE Key Laboratory of Functional Small Organic Molecule College of Chemistry and Chemical Engineering, Jiangxi Normal University, 330022 Nanchang, China. These authors contributed equally: Yi-Rong Wang, Qing Huang and Chun-Ting He. Correspondence and requests for materials should be addressed to Y.-Q.L. (email: yqlan@njnu.edu.cn)

The anthropic excessive emission of carbon dioxide ($CO_2$) has resulted in serious climate change and environmental crisis, such as global warming, acid rain, and rising of sea level, etc[1]. Powerful methods like storage and conversion of $CO_2$ have been developed to reduce atmospheric $CO_2$ level[2–4]. In contrast, conversion of $CO_2$ is regarded as more promising way, due to which can make $CO_2$ transform into serviceable high-valued energy products or chemicals (e.g., CO, $CH_4$, HCOOH and $C_2H_5OH$, etc.), that would simultaneously mitigate the greenhouse effect and energy crisis[5–7]. At present, the most representative strategy for implementing this conversion process is electrocatalytic $CO_2$ reduction reaction ($CO_2$RR), because of its characteristic advantages of easy processing, simple devices, and low energy consumption[8–13]. Nevertheless, the electrochemical conversion of $CO_2$ still needs to conquer considerable kinetic barrier required by the intrinsically thermodynamic stability and low electron affinity of the $CO_2$ molecule, and the accompanying competitive $H_2$ evolution is always hindering the generation of aiming product, resulting in hard to discover suitable electro-catalysts to afford high $CO_2$RR efficiency and selectivity[14].

During past decades, researchers have explored different kinds of materials for $CO_2$RR electrocatalysis, such as metals[8,15], transition metal oxides[9,16], transition metal chalcogenides[17,18], metal-free 2D materials[14], and metal-organic frameworks (MOFs)[19–23], etc[24–27], among which MOFs are regarded as ascendant $CO_2$RR electrocatalysts owing to their structural merits commonly involve in open Lewis acid metal sites in favor of adsorbing $CO_2$ molecule, as well as well-defined catalytic environments suitable for mechanism explanation[28–31]. However, it is a pity that their poor electrical conductivity and electron-donating capability are always the major constraints for MOFs being as efficient electrocatalysts, considering the accomplishment of necessary multiple electron transfer process for obtaining any reductive products in $CO_2$RR. Consequently, integrating electron-rich unit, electron mobility and active component related to specifically electrocatalytic $CO_2$ reduction product, into MOF will likely to be an effective approach to improve the selectivity and efficiency of $CO_2$RR.

As a proof-of-concept, we decided that assembling reductive polyoxometalates (POMs) and metalloporphyrins to construct MOFs so as to verify the feasibility of this inference[32,33]. The corresponding considerations are listed as follows. Firstly, reductive POMs mainly composed of low valent metal ions, such as Zn-ε-Keggin cluster ({ε-$PMo_8{}^VMo_4{}^{VI}O_{40}Zn_4$})[34,35], including eight $Mo^V$ atoms), are usually electron-rich aggregates[36] and can easily offer electrons when triggered by redox reaction or bias stimulus; secondly, metalloporphyrin, where inherent macrocycle conjugated π-electron system is very beneficial for electron mobility, has proven to be advantageous to produce CO in electrocatalytic $CO_2$RR[19,37–39]; thirdly, the connection of POM and metalloporphyrin will presumably create an oriented electron transportation pathway under the motivation of electric field, abundant electrons flowing from POM cluster to metalloporphyrin motif can guarantee and facilitate the fulfillment of multiple electron migration process of $CO_2$RR; and finally, polyoxometalate-based MOFs always performing good structural robustness and chemical stability that is essential to the durability test of $CO_2$RR. From the above we deduce that reductive polyoxometalate-metalloporphyrin organic frameworks (PMOFs) will probably be promising candidates to enhance the efficiency and selectivity of $CO_2$RR, while the assembly of this type of structures is still extremely challenging[40].

Herein, we synthesize a series of stable PMOFs constructed by tetrakis[4-carboxyphenyl]-porphyrin-M (M-TCPP) linkers and reductive Zn-ε-Keggin clusters through an in situ hydrothermal method with similar formulae of [$PMo^V{}_8Mo^{VI}{}_4O_{35}(OH)_5Zn_4$]₂[M-TCPP][$2H_2O$][1.5TBAOH] (M = Fe, Co, Ni, and Zn). The synergistic combination of Zn-ε-Keggin and M-TCPP in these PMOFs can serve as the role of gathering electron donating, electron migration, and electrocatalytic active component in the $CO_2$RR. The electroreduction performances of diverse transition metal (i.e., Co, Fe, Ni, and Zn) based PMOFs are tested and relative results are discussed in the single-crystal level. As expected, these PMOFs show excellent performances in electrocatalytic $CO_2$RR, especially for Co-PMOF, which is able to selectively convert $CO_2$ to CO with a superior faradaic efficiency of 99% (highest in reported MOFs), a high TOF of 1656 h$^{-1}$ (−0.8 V) and excellent stability for more than 36 h. It is worth noting that this is the first case to explore polyoxometalate-based MOF in the application of electrochemical $CO_2$RR.

## Results

**Structure and characterization of M-PMOFs.** M-PMOF (i.e., Co-PMOF, Fe-PMOF, Ni-PMOF, and Zn-PMOF) samples were synthesized under similar hydrothermal conditions. A mixture of M-TCPP, $Na_2MoO_4{\cdot}2H_2O$, $H_3PO_3$ and zinc chloride was treated at 180 °C for 72 h and thus-obtained crystals were washed and measured. Single-crystal X-ray diffraction studies reveal that M-PMOF crystals have iso-reticular structures and are crystallized in orthorhombic **Fmmm** space group (Fig. 1). In the structure, Zn-ε-Keggin and M-TCPP present tetrahedral and quadrilateral connection modes, respectively. One Zn-ε-Keggin cluster with four Zn (II) centers connects with adjacent Zn-ε-Keggin clusters with two Zn-O bonds to give cluster chain. Chains are further connected by M-TCPP ligands to provide a three-dimensional framework with two-fold interpenetrated **mog** topology (Supplementary Fig. 1). Typically, in the structure, four Zn-ε-Keggin chains (adjacent distance is about 15 Å) are linked by M-TCPP ligands through the coordination of Zn (II) and carboxylic groups. The distance between two nearest and parallel M-TCPP units in uninterpenetrated network is about 17 Å (Fig. 1). The interpenetrated topology might endow these structures with high chemical, thermal, or catalysis stability and the direct connection of Zn-ε-Keggin with M-TCPP ligands in a structure would presumably create an oriented electron transportation pathway that might be beneficial for efficient charge transfer.

To investigate their stability, M-PMOF samples are successfully prepared and certified by the powder X-ray diffraction (PXRD) tests (Supplementary Figs. 3–6). The thermal stability of M-PMOF samples is studied by thermogravimetric analysis (TGA) in $O_2$ atmosphere with a heating rate of 10 °C min$^{-1}$. Under $O_2$ atmosphere, all these M-PMOF samples can be stable at temperatures higher than 200 ºC. Taking Co-PMOF for example, about 4% mass loss at temperature range from 0 to 100 ºC is attributed to the loss of guest molecules. After 200 °C, the framework of Co-PMOF starts to collapse and ends at about 500 °C (Supplementary Fig. 8). To further certify the thermal stability, PXRD patterns of M-PMOF samples treated at 200 °C in the presence of ultrapure $O_2$ were tested. As shown in Supplementary Fig. 9, the PXRD patterns of M-PMOF samples still agree well with the simulated ones. Furthermore, M-PMOF samples show high chemical stability. All the M-PMOF samples enable to be stable in $KHCO_3$ aqueous (0.5 M) for more than 24 h, guaranteeing the further investigation of electroreduction performance (Supplementary Figs. 10–13). Moreover, after 24 h test, all of them can maintain their structures at pH range from 5 to 11 proved by PXRD tests, IR, and mapping test, respectively (Supplementary Figs. 14, 15). The high chemical and thermal stability of M-PMOF samples set fundamental basis for further applications.

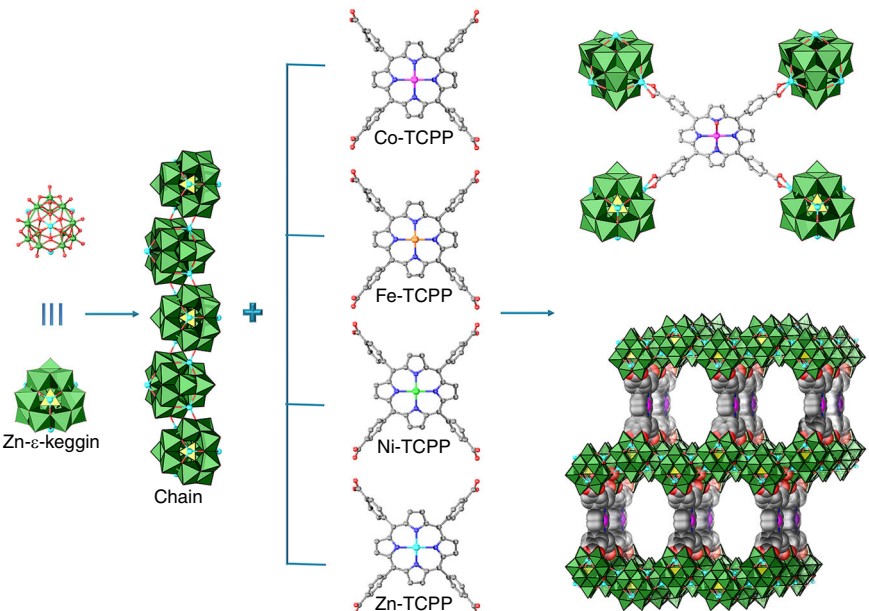

**Fig. 1** Schematic illustration of the structures of M-PMOFs (M = Co, Fe, Ni, Zn). M-PMOF was constructed by the 4-connected TCPP linkers and zigzag POM chains

**The electrocatalytic performance of M-PMOFs**. The electrochemical $CO_2RR$ performance is evaluated in an airtight three electrodes electrochemical H-type cell (the compartments are separated by a proton exchange membrane (Nafion®212)). All the M-PMOFs (i.e., Co-PMOF, Fe-PMOF, Ni-PMOF, and Zn-PMOF) samples are packed in the cells and tested. In this work, all the potentials are measured vs. Ag/AgCl electrode and the results are reported vs. reversible hydrogen electrode (RHE).

Linear sweep voltammetry (LSV) curves (without *iR* compensation) show that Co-PMOF has a small onset potential of −0.35 V and is much more positive than that of Fe-PMOF (−0.53 V), Ni-PMOF (−0.58 V), and Zn-PMOF (−0.60 V) in $CO_2$-saturated 0.5 M $KHCO_3$ (pH = 7.2) solution (Fig. 2a). Co-PMOF exhibits a small overpotential of 0.24 V and the relative equilibrium potential of $CO_2$/CO is −0.11 V. Besides, Co-PMOF presents a total current density of 38.9 mA cm$^{-2}$ at −1.1 V and is superior to Fe-PMOF (25.1 mA cm$^{-2}$), Ni-PMOF(20.02 mA cm$^{-2}$), and Zn-PMOF (16 mA cm$^{-2}$). Due to its excellent performances, this work will mainly focus on Co-PMOF to discuss the possible relations between its properties and well-defined structure.

Combing POMs and M-TCPP as building units, the high electrochemical activity of Co-PMOF is supposed to come from $CO_2RR$. To prove it, relative tests are conducted both in $N_2$ and $CO_2$ saturated $KHCO_3$ solution. As expected, Co-PMOF exhibits a more negative onset potential and a smaller current density in $N_2$-saturated $KHCO_3$ solution than that in $CO_2$-staurated $KHCO_3$ solution (Supplementary Fig. 16). Furthermore, the GC analyses show that $H_2$ and CO are the primary reduction products and there is no liquid product detected by $^1H$ nuclear magnetic resonance spectroscopy (Supplementary Fig. 17). To further evaluate the selectivity of M-PMOF samples for $CO_2RR$, the corresponding FE for CO and $H_2$ (denoted as $FE_{CO}$ and $FE_{H2}$) are calculated over the entire potential range (Fig. 2b and Supplementary Fig. 18). Taking Co-PMOF for example, the initial formation of CO is detected by GC at the potential of −0.35 V with a small CO partial current density of 0.39 mA cm$^{-2}$. With the increase of potential, the $FE_{CO}$ continuously increases and reaches up to a maximum value of 98.7% at −0.8 V and keeps higher than 94% even at −1.0 V (Supplementary Fig. 19). The

high $FE_{CO}$ over such as wide potential range (from −0.8 V to −1.0 V) outperforms many reported materials (Supplementary Table 1)[41–43]. Noteworthy, the maximum $FE_{CO}$ value (98.7%) of Co-PMOF is higher than other M-PMOFs (i.e., Fe-PMOF, 28.8%; Ni-PMOF, 18.5%, and Zn-PMOF, 1.2%, respectively) and is highest among reported MOFs (Supplementary Table 1)[36,37]. In addition, the same electrochemical experiment of Co-PMOF is carried out in $N_2$-saturated 0.5 M $KHCO_3$ to reveal the source of CO and there is no CO detected under this condition (Supplementary Fig. 21). Besides, an isotopic experiment that using $^{13}CO_2$ as substrate was performed under identical reaction conditions to reveal the carbon source of CO. The products were analyzed by gas chromatography and mass spectra. As shown in Supplementary Fig. 22, the peak at $m/z = 29$ is assigned to $^{13}CO$, indicating that the carbon source of CO indeed derives from the $CO_2$ used.

To further reveal the efficiency of the remarkable performances of Co-PMOF, partial current densities of CO and $H_2$ at different potentials are detected (Fig. 2c and Supplementary Fig. 23). Co-PMOF gives a partial CO current density of 18.08 mA cm$^{-2}$ at −0.8 V. This value is 30 times larger than that of Fe-PMOF (0.27 mA cm$^{-2}$), Ni-PMOF (0.47 mA cm$^{-2}$), and Zn-PMOF (0.02 mA cm$^{-2}$). Besides, TOF of Co-PMOF is calculated to be 1656 h$^{-1}$ at −0.8 V and the TON values of Co-PMOF following the change of time was shown in Supplementary Fig. 24[37]. To the best of our knowledge, this catalytic behavior of Co-PMOF outperforms most of MOF catalysts and is one of the best reported materials (Supplementary Table 1)[43,44].

To elucidate the dynamics activity of Co-PMOF for electrochemical $CO_2RR$, Tafel slopes are calculated based on Tafel equation ($\eta = b \log j + a$, where $\eta$ is the overpotential, $j$ is the current density and b is the Tafel slope) and presented in Fig. 2d. Remarkably, the Tafel slope for Co-PMOF is 98 mV dec$^{-1}$, which is much smaller than that of Fe-PMOF (211 mV dec$^{-1}$), Ni-PMOF (675 mV dec$^{-1}$), and Zn-PMOF (206 mV dec$^{-1}$). This indicates the favorable kinetics of Co-PMOF for the formation of CO, which might be ascribed to the more efficient charge transfer and larger active surface in the catalytic process. To support the hypothesis, electrochemical double-layer capacitance ($C_{dl}$) is calculated to estimate the electrochemical active surface area

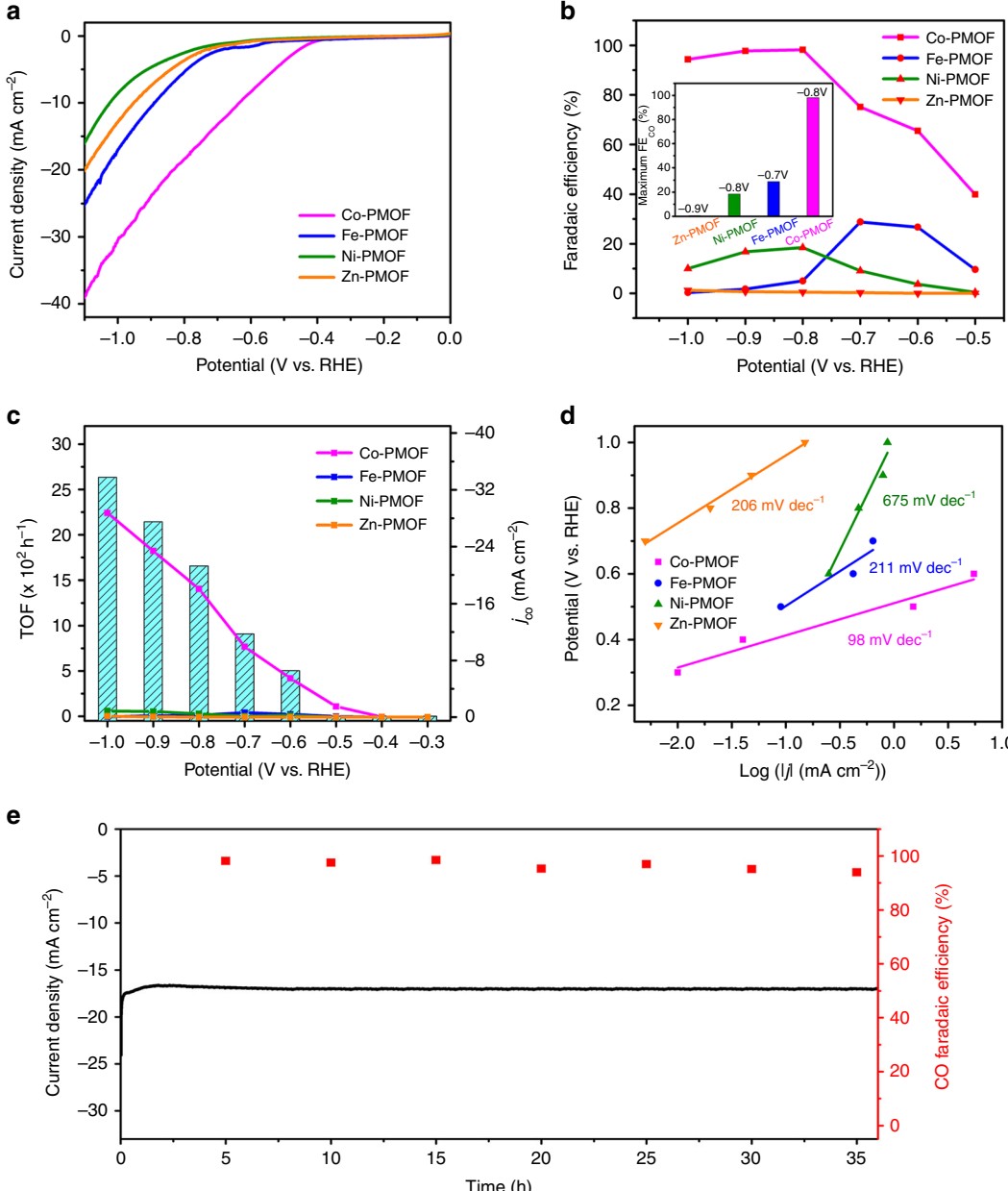

**Fig. 2** Electrocatalytic performances of M-PMOFs. **a** Linear sweep voltammetric curves. **b** Faradaic efficiencies for CO. **c** Partial CO current density and TOFs. **d** Tafel plots. **e** Durability test of Co-PMOF at the potential of −0.8 V vs. RHE. These tests are performed in 0.5 M KHCO₃ electrolyte and at the scan rate of 5 mV s⁻¹

(ECSA) to further discuss the potential influence factors (Supplementary Fig. 25)[45]. The result show that Co-PMOF indeed exhibits larger $C_{dl}$ value (12.17 mF cm⁻²) than that of Fe-PMOF (10.26 mF cm⁻²), Ni-PMOF (10.16 mF cm⁻²), and Zn-PMOF (9.83 mF cm⁻²), which enable to provide more active sites in electrocatalyst to contact the electrolyte for being beneficial to increase the reaction speed of electrochemical CO₂RR.

Moreover, electrochemical impedance spectroscopy (EIS) measurement is carried out to probe the electrocatalytic kinetics on the electrode/electrolyte surface. The Nyquist plots demonstrate that Co-PMOF has much smaller charge transfer resistance (9.83 Ω) than Fe-PMOF (10.26 Ω), Ni-PMOF (10.70 Ω), and Zn-PMOF (12.17 Ω) during the electrochemical CO₂RR (Supplementary Fig. 26). This indicates Co-PMOF can provide faster

electron transfer from the catalyst surface to the reactant (i.e., CO₂) in intermediate (HCOO* and CO*) generation, eventually resulting in largely enhanced activity and selectivity. This conclusion will be further verified by the theoretical calculations and relative contents are discussed in the mechanism part.

Stability is a key factor to evaluate the durability properties of catalysts. To evaluate the CO₂RR stability of Co-PMOF, long-time durability test is performed with chronoamperometric test at a fixed overpotential of 0.69 V in 0.5 M KHCO₃ solution (Fig. 2e). After 36 h, negligible decay in activity and FE_CO is detected (the gaseous product is analyzed by GC every 5 h). During the process, the corresponding FE_CO can be retained at values more than 90% and the current density is about 17 mA cm⁻² over the entire experiment. This stability measurement reveals that Co-PMOF is

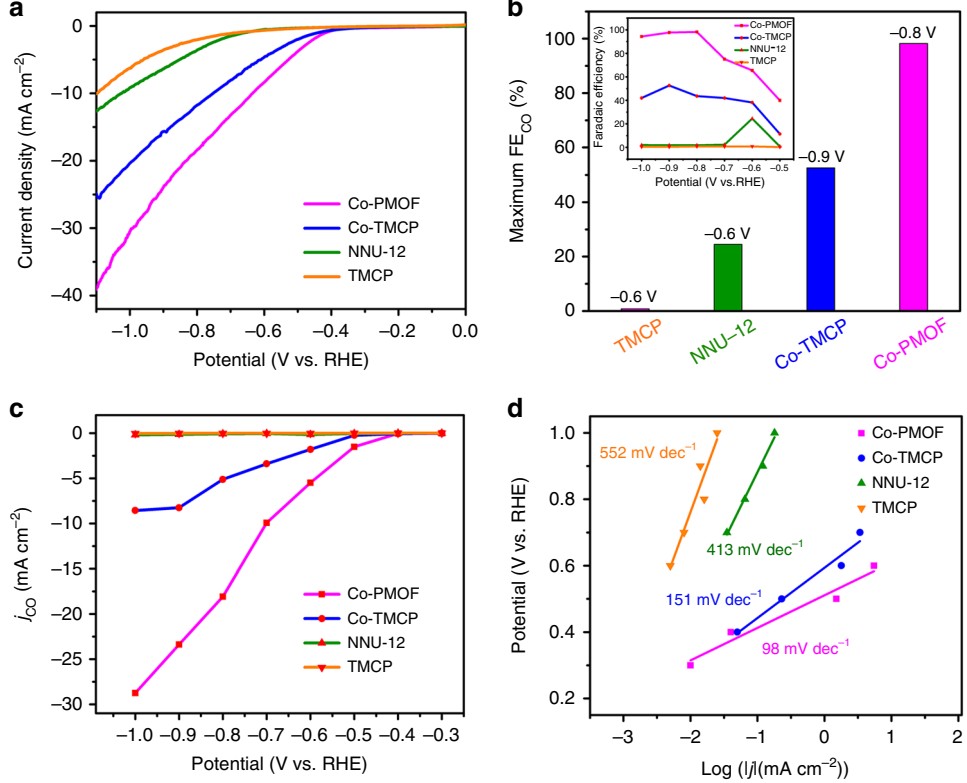

**Fig. 3** Electrocatalytic performances of contrast samples for Co-PMOF. **a** Linear sweep voltammetric curves. **b** Maximum FE$_{CO}$ of TMCP ($-0.6$ V), NNU$-12$ ($-0.6$ V), Co-TMCP ($-0.9$ V) and Co-PMOF ($-0.8$ V). **c** Partial CO current density. **d** Tafel plots

highly stable electrocatalyst, which has much potential to be used in efficient electrochemical CO$_2$RR (Supplementary Figs. 27, 38).

## Discussion

To reveal the catalytic mechanism, a series of contrast samples are prepared and tested. The electrochemical CO$_2$RR of contrast samples are measured adopting the same testing methods. To evaluate the impact of Co-TCPP in electrochemical CO$_2$RR, NNU-12 $\{[PMo_8^VMo_4^{VI}O_{36}(OH)_4Zn_4][BCPT]_2 \cdot xGuests\}$ is prepared[46]. NNU-12 with ***dia*** topology is constructed by Zn-ε-Keggin and BCPT$^{2-}$ linker, which contains the same Zn-ε-Keggin unit as Co-PMOF while the ligand is different. NNU-12 shows poor current density (3.63 mA cm$^{-2}$ at $-0.8$ V), low FE$_{CO}$ (1.80% at $-0.8$ V), high onset potential ($-0.6$ V) with a Tafel slope of 413 mV dec$^{-1}$, and has negligible CO$_2$ reduction activity (Fig. 3, Supplementary Figs. 28, 30). This indicates the high CO$_2$ reduction activity of Co-PMOF originates from Co-TCPP. In the process, Co-TCPP might act as electrocatalytic center and redox-hopping-based conduit, which is beneficial for charge transfer.

The influence of reductive POM in the electrochemical CO$_2$RR is also investigated. Tetramethyl 5, 10, 15, 20-tetrakis[4-carboxyphenyl]-porphyrin-Co (Co-TMCP) is prepared and shows a small onset potential of $-0.4$ V, high current density (11.60 mA cm$^{-2}$ at $-0.8$ V) with a Tafel slope of 151 mV dec$^{-1}$ (Supplementary Fig. 29 and Fig. 3). In contrast, Co-TMCP exhibits poorer CO$_2$ reduction performance than Co-PMOF, which might be attributed to the lower proton and electron transfer efficiency (Fig. 3 and Supplementary Fig. 30). Zn-ε-Keggin with strong reducibility might provide additional electrons and facilitate the charge transfer for Co-PMOF in electrochemical CO$_2$RR.

In addition, tetramethyl 5, 10, 15, 20-tetrakis[4-carboxyphenyl]-porphyrin (TMCP) is prepared to study the effect of Co in the reduction process. TMCP shows poorer current density

(2.06 mA cm$^{-2}$ at $-0.8$ V), lower FE$_{CO}$ (0.77% at $-0.8$ V), higher onset potential ($-0.67$ V) with a Tafel slope of 552 mV dec$^{-1}$ and has no CO$_2$RR activity than Co-PMOF (Fig. 3, Supplementary Figs. 29, 30), implying Co is the electrochemically active site in Co-PMOF. To further support the results, ECSA and EIS measurements are conducted and relative results are calculated. These results give similar performance tendency in Co-PMOF as discussed above (Supplementary Fig. 34 and 35). Moreover, MOF-525(Co) was prepared as relevant comparison to further support the catalytic mechanism of M-PMOF (Supplementary Fig. 31). MOF-525(Co) has similar ligand as Co-PMOF and is constructed with Zr$_6$O$_4$(OH)$_4$ unit without POM[47]. MOF-525(Co) is successfully prepared and certified by the PXRD tests (Supplementary Fig. 32). The electrochemical CO$_2$RR of MOF-525(Co) is measured adopting the same testing methods. As shown in Supplementary Fig. 33, MOF-525(Co) shows lower FE$_{CO}$ (47.9% at $-0.8$ V) than Co-PMOF (98.7% at $-0.8$ V), which might be attributed to the poorer proton and electron transfer efficiency. This indicates POM actually acts as electron-rich aggregates in the catalytic mechanism of M-PMOF.

To understand the highly active as well as selective reaction mechanism of Co-PMOF, DFT calculations are performed (Supplementary note 2). Generally, the electroreduction of CO$_2$ to CO includes three elementary reactions, the formation of *COOH and *CO with one electron transfer for each of them, and the finally CO desorption process. The asterisk (*) represents the surface active sites for adsorption and reaction. From the calculated free energy diagrams shown in Fig. 4a, the rate-determining steps (RDS) for CO$_2$ reduction on single Zn-ε-Keggin (POM) and Co-TCPP are the formation of adsorbed intermediates *COOH and *CO with rather high free energies of $\Delta G_1 = 0.96$ eV and $\Delta G_2 = 0.53$ eV, respectively. Expectedly, when assembling the Zn-ε-Keggin and Co-TCPP together, the final compound of Co-PMOF possesses remarkably reduced $\Delta G_1$ and $\Delta G_2$, and

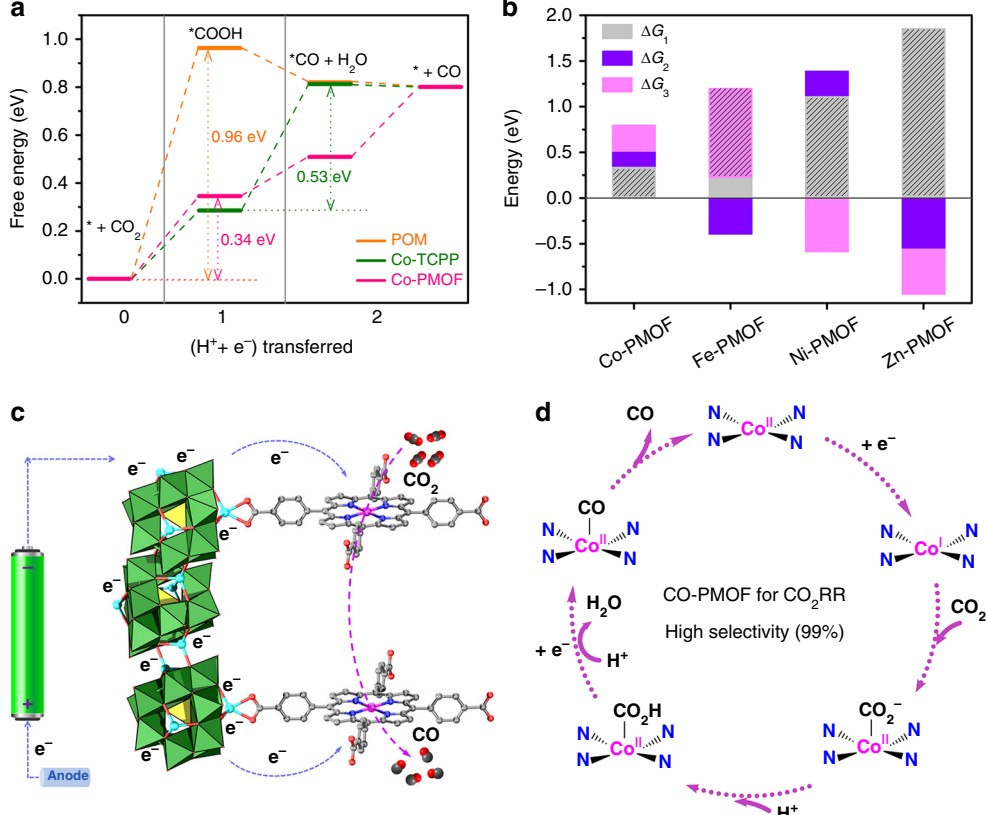

**Fig. 4** The DFT calculation and proposed reaction mechanism. **a** The free energy diagrams of $CO_2$ reduction to CO for POM (orange), Co-TCPP (green), and Co-PMOF (pink). The rate-determining step and the corresponding free energy on each material are indicated (Note that asterisk represents a surface active site for reaction). **b** Comparison of the free energy of each elementary reaction ($\Delta G_1$, $\Delta G_2$, and $\Delta G_3$ represent the free energy of *COOH formation, *CO formation, and CO desorption process, respectively) in $CO_2$RR for Co-PMOF, Fe-PMOF, Ni-PMOF, and Zn-PMOF, respectively. **c, d** Proposed mechanistic scheme for the $CO_2$RR on Co-PMOF (POM as electron-donating group, porphyrins as charge transfer ligand and transition metals as electron-collecting group)

particularly the RDS of *COOH formation decrease to a much smaller free energy of $\Delta G_1 = 0.34$ eV, being consistent with the high electroreduction activity and selectivity of this PMOF. These results indicate not only the favorable active site is the Co in Co-TCPP instead of POM, but also the efficiently synergistic electron modulation of POM and the porphyrin metal center. Then, the reaction pathways are further calculated by replacing the Co in porphyrin center with Fe, Ni, and Zn (Fig. 4b and Supplementary Fig. 36). For Fe-PMOF, although the energy of *COOH and *CO formations have decreased when compared to that of Co-PMOF, the desorption of CO become harder ($\Delta G_3 = 0.98$ eV), which may be ascribed to the higher affinity of Fe. For the case of Ni and Zn, the RDS both are the formation of *COOH with much higher free energies of 1.12 and 1.86 eV, respectively. This indicates the initial proton-coupled electron transfer to $CO_2$ forming an adsorbed *COOH intermediate is sluggish in these systems. In a word, the sequence of the reaction energies on different metal centers is well consistent with our experiments, which further confirms the high activity of the Co modified compounds and the modulation effect in the M-PMOF systems.

Based on the experiment results and theoretical calculations, we elaborate the reduction processes from $CO_2$ to CO on Co-PMOF and the possible mechanism is summarized in Fig. 4d and Supplementary note 3. During the electrochemical process, Co centers are reduced from Co(II) to Co(I) proved by the cyclic voltammetry (CV) tests (Supplementary Fig. 37), which has been detected in many Co-centered macrocycles[11,29,37,48]. As proposed

in the mechanism, Zn-ε-Keggin initially traps the electron from the electrodes and transfers it to the Co center (Fig. 4c). Meanwhile, the Co center is reduced from Co(II) to Co(I) during the process. Then Co(I) interacts with carbon monoxide to give Co(II)*COOH intermediate coupling with the proton-electron transfer. Subsequently, Co(II)*COOH convert to Co(II)*CO. Finally, CO is desorbed and produced (Fig. 4d).

In summary, we have designed and synthesized a series of stable PMOFs by applying reductive Zn-ε-Keggin cluster and metalloporphyrin as building block and linker, respectively. The direct communication of reductive POM unit and metalloporphyrin under the exertion of the electric field results in an oriented electronic transportation channel that is extremely conductive to the accomplishment of multiple electrons transfer process in electrocatalytic $CO_2$RR. Obviously, these PMOFs show excellent electrochemical $CO_2$ reduction performances at last. In particular for Co-PMOF, the best one of all target compounds, enables to selectively convert $CO_2$ to CO with a remarkable faradaic efficiency of 99% (highest in reported MOFs), a high TOF of 1656 h$^{-1}$ ($-0.8$ V) and excellent stability for more than 36 h. Specially, DFT calculation demonstrates the superior catalysis performances of Co-PMOF than other transition metals (i.e., Fe, Ni, and Zn) and further reveals the synergistic effect of reductive POM and Co-porphyrin. This strategy opens great perspectives in designing novel and efficient $CO_2$RR electrocatalysts and might shed light on the exploration of powerful protocols to address $CO_2$ problems.

## Methods

**Syntheses of various M-TMCPs.** 5, 10, 15, 20-Tetrakis(4-methoxycarbonylphenyl)porphyrin (TMCP) (1.7 g, 2.0 mmol), CoCl$_2$·6H$_2$O (6.1 g, 25.6 mmol), and DMF (200 mL) were mixed and refluxed for 12 h. A large amount of water was added after cooling down to room temperature. The as-synthesized material was filtered and washed with water (600 ml) for six times. The fuchsia solid sample was obtained (yield 90% based on TMCP). The preparation processes of Fe-TMCP, Ni-TMCP, and Zn-TMCP were similar to Co-TMCP except that CoCl$_2$·6H$_2$O (6.1 g, 25.6 mmol) was replaced by FeCl$_2$·4H$_2$O (5.0 g, 25.6 mmol), NiCl$_2$·6H$_2$O (6.2 g, 25.6 mmol), and ZnCl$_2$ (3.5 g, 25.6 mmol), respectively[49].

**Syntheses of various M-TCPPs.** The obtained ester, MeOH (50 mL) and THF (50 mL) were mixed and stirred. A volume of 50 mL KOH solution (5.2 g, 93.9 mmol KOH was dissolved in 50 mL H$_2$O) was added into the mixture and was refluxed for one night. After reaction, the organic solvents were evaporated. To make the solid fully dissolve, moderate water was added to the mixed solution and heated. 2 M HCl was used to acidify the obtained fuchsia homogeneous solution until no further fuchsia precipitate was generated. The solid was collected and washed with deionized water until the pH of filtrate reaches about 5. Finally, the solid sample was dried in vacuum at 60 °C. The colors of Co-TCPP, Fe-TCPP, and Ni-TCPP were red, brown, and crimson, respectively[49].

**Syntheses of various M-PMOFs.** Taking Co-PMOF for example, a mixture of H$_3$PO$_3$ (10 mg, 0.125 mmol), zinc chloride (68 mg, 0.50 mmol), Na$_2$MoO$_4$·2H$_2$O (310 mg, 1.28 mmol), and tetrabutylammonium hydroxide (TBAOH) solution (250 µL, 10 wt % in water) was added into 3.5 mL H$_2$O and stirred for 15 min. 2 mol L$^{-1}$ HCl solution was used to adjust the pH of the mixture solution until the pH reached about 5. Then, 5,10,15,20-tetrakis(4-carboxyphenyl)porphyrinato-Co (Co-TCPP) (132 mg), Mo powder 99.99% (25 mg), and dimethylacetamide solvent (250 µL) were added into the mixed solution and stirred in a 15 mL Teflon-lined stainless steel autoclave and heated at 180 °C for 72 h. After temperature cooled down to room temperature with a rate of 15 °C h$^{-1}$, the crystals were washed with water and collected (yield was about 75% based on Co-TCPP). The preparation processes of Ni-PMOF and Fe-PMOF were similar to Co-PMOF except that Co-TCPP (132 mg) was replaced by Ni-TCPP (132 mg) and Fe-TCPP (33 mg), respectively. It is worthy to note that the preparation process of Zn-PMOF was similar to Co-PMOF except that Co-TCPP (132 mg) is replaced by TCPP (118 mg) (Supplementary Fig. 2). The IR peaks are listed as follows (Supplementary Fig. 7, KBr pellets, ν cm$^{-1}$): Co-PMOF: 3427 (s), 3122 (w), 2958 (w), 1699 (w), 1604 (s), 1400 (s), 1040 (m), 1271 (w), 1172 (w), 1027 (m), 941 (s), 869 (w), 796 (s), 713 (m), 594 (w), 495 (w); Fe-PMOF: 3446 (s), 3211 (w), 1701 (w), 1602 (s), 1404 (s), 1271 (w), 1043 (m), 931 (s), 864 (w), 783 (m), 714 (m), 595 (w), 489 (w); Ni-PMOF: 3436 (s), 3214 (w), 1704 (w), 1606 (s), 1399 (s), 1261 (w), 1041 (m), 930 (s), 861 (w), 785 (m), 712 (m), 597 (w), 490 (w) and Zn-PMOF: 3431 (s), 3188 (w), 1703 (w), 1600 (s), 1401 (s), 1272 (w), 1038 (m), 937 (s), 865 (w), 789 (m), 711 (m), 596 (w), 492 (w). For the formulae of M-PMOF, all of these M-PMOF are iso-reticular in structure and only have minor difference in centered metal ions. Their structures have been well-defined in the SXRD tests. Based on the elemental analyses and SXRD tests, the calculated formulae was [PMo$^V_8$Mo$^{VI}_4$O$_{35}$(OH)$_5$Zn$_4$]$_2$[M-TCPP][2H$_2$O][1.5TBAOH] (M = Fe, Co, Ni, and Zn). This result was further supported by TGA test (Supplementary Fig. 8).

**Synthesis of MOF-525.** ZrCl$_4$·8H$_2$O (12.5 mg, 0.037 mmol) and DMF (10 mL) were sonicated for 30 min. Then, tetrakis(4-carboxyphenyl)porphyrin (2.5 mg, 0.037 mmol) was added into the solution. After sonication for 10 min, acetic acid (2.5 mL) was added. The solution was placed in a 20 mL scintillation vial and heated at 65 °C for 3 days. The microcrystalline powder was filtered and washed with DMF, acetone for several times. Finally, the powder was dried in vacuum at 120 °C for 48 h[47].

**Synthesis of MOF-525(Co).** CoCl$_2$·6H$_2$O (100 mg, 0.62 mmol), MOF-525 (50 mg), and DMF (10 mL) were mixed and heated at 100 °C for 18 h. The microcrystalline powder were collected by filtration and washed with DMF, acetone for several times. Finally, the powder was dried in vacuum at 120 °C for 48 h[47].

**Characterizations and instruments.** Powder X-Ray diffraction (PXRD) patterns of samples were carried out on a D/max 2500 VL/PC diffractometer (Japan) equipped with Cu Kα radiation (λ = 1.54060 Å). FTIR spectra were tested on a Bruker Tensor 27 FT/IR spectrophotometer. Thermogravimetric analyses of crystal samples were measured on a Netzsch STA449F3 analyzer under the oxygen atmosphere with a heating rate of 10 °C min$^{-1}$. Single-crystal XRD data of Co-PMOF, Fe-PMOF, Ni-PMOF, and Zn-PMOF were measured on a Bruker APEXII CCD diffractometer with graphite-monochromated Mo Kα radiation (λ = 0.71073 Å) at 296 K. All the structures were solved by direct method using SHELXT program and refined on an Olex$^2$ software. Multi-scan technique was used to absorption corrections. The TBA$^+$ ion could not be detected in the structure and some other guest molecules were defined as disorder. The detailed crystallographic information of these crystals was listed in Supplementary Table 2. The CCDC number of Co-PMOF, Fe-PMOF, Ni-PMOF, and Zn-PMOF (NNU-13) were

1832596 [https://doi.org/10.5072/ccdc.csd.cc1zjbmg], 1832597 [https://doi.org/10.5072/ccdc.csd.cc1zjbnh], 1832598 [https://doi.org/10.5072/ccdc.csd.cc1zjbpj], and 1181860 [https://doi.org/10.5072/ccdc.csd.cc1ytd38] respectively.

**Electrochemical measurements.** All electrocatalysis tests of the catalysts were performed at ambient environment on the electrochemical workstation (Bio-Logic) in a standard three-electrode configuration in 0.5 M KHCO$_3$ (pH = 7.2). Pt wire and Ag/AgCl electrode were used as the counter and reference electrode, respectively. The working electrode was modified carbon cloth (1 cm × 1 cm). The experiment was performed in an airtight electrochemical H-type cell with a catalyst-modified carbon cloth electrode (denoted as CCE, 1 cm × 2 cm) as the work electrode.

Given the poor intrinsic electrical conductivity of MOFs, acetylene black (AB) was introduced to mix with the as-synthesized MOFs to improve the conductivity. Nafion solution was introduced as a kind of MOF dispersion solution generally applied in many reported works, which can form a homogeneous ink with MOF and further help to attach onto the surface of carbon cloth. The pure carbon cloth has no CO$_2$RR activity (Supplementary Fig. 20). The preparation of the CCE working electrode was as follows. A volume of 10 mg electrocatalyst, 10 mg acetylene black (AB), and 1 mL 0.5% Nafion solution were grounded to form uniform catalyst ink. After sonication for 30 min, the ink was dropped directly onto a carbon cloth (1 cm × 1 cm) with a catalyst loading density of ~1 mg cm$^{-2}$ and dried. In the H-type cell, two compartments were separated by an exchange membrane (Nafion®212).

During the CO$_2$ reduction experiments, the polarization curves were performed by LSV mode at a scan rate of 5 mV s$^{-1}$. Initially, polarization curves for the modified electrode were recorded under an inert N$_2$ (gas) atmosphere. After that, the solution was bubbled with CO$_2$ (99.999%) for at least 30 min to make the aqueous solution saturated and then the electrocatalytic CO$_2$RR was conducted (Supplementary note 1). Potential was measured vs. Ag/AgCl electrode and the results were reported vs. reversible hydrogen electrode based on the Nernst equation: E (vs. RHE) = E (vs. Ag/AgCl) + 0.1989 V + 0.059 × pH.

EIS measurement was carried out by applying an AC voltage with 10 mV amplitude in a frequency range from 1000 kHz to 100 mHz at a overpotential of −0.8 V (vs. RHE). To estimate the ECSA, cyclic voltammograms (CV) were tested by measuring double-layer capacitance (C$_{dl}$) under the potential window of −0.45 V to −0.55 V (vs. Ag/AgCl) with various scan rates from 10 to 100 mV s$^{-1}$. All the LSV curves were presented without iR compensation.

## Data availability

The data that support the findings of this study are available from the corresponding author upon reasonable request.

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

## Acknowledgements

This work was financially supported by NSFC (No. 21622104, 21471080, and 21701085), the NSF of Jiangsu Province of China (No. BK20171032), the Natural Science Research of Jiangsu Higher Education Institutions of China (No. 17KJB150025), Priority Academic Program Development of Jiangsu Higher Education Institutions, and the Foundation of Jiangsu Collaborative Innovation Center of Biomedical Functional Materials.

## Author contributions

Y.-Q.L., Y.-R.W., and Q.H., conceived the idea. Y.-R.W., Q.H., Y.-F.C., and J.L. designed the experiments and collected and analyzed the data. C.-T.H. accomplished the theoretical calculation. F.-C.S and J.L. assisted with the experiments and characterizations. Y.-R.W. wrote the manuscript. All authors discussed the results and commented on the manuscript.

## Additional information

**Competing interests:** The authors declare no competing interests.

