## [Peer Review File · Nature Communications]

Reviewers' comments:

Reviewer #1 (Remarks to the Author):

Lan and coworkers have synthesized a series of novel polyoxometalate-metalloporphyrin organic frameworks (PMOFs) crystals and explored their applications in CO₂ reduction. By integration of Zn- ϵ -Keggin cluster and metalloporphyrin the electric conductivity and electron donating capability of the fabricated MOFs are substantially strengthened. Among these PMOFs, Co-PMOF exhibits remarkable faradaic efficiency up to 99% for CO and long-term stability. Further, the mechanisms of catalysis processes are also discussed and revealed by DFT calculations. The results reported here are quite interesting and inspiring. Therefore, I recommend that this work for publication after minor revision before the following questions are fully addressed in order to further improve the quality of the nice work.

One of the major problem to hinder the MOFs as electrocatalysts is their stability. The authors claimed that the PMOFs, for example, Co-PMOF exhibited excellent electrocatalytic stability (> 36 h) and provided solid evidences of the Co-PMOFs in solutions with various pH values, but before electrocatalytic reaction. I am very curious to know whether the authors have characterized Co-PMOF after electrocatalysis. Therefore, I would suggest the authors, at least, should provide the XRD pattern of Co-PMOF after 36 h.

Concerning the mechanism proposed in Figure 4d, Co(I) species are involved during the electrocatalytic process. If the authors can provide direct evidences about the existence of the Co(I), such as EPR experiment, the proposed mechanism will be more solid.

Other suggestions:

1. The title "Oriented Electron Transmission in POM-Metalloporphyrin Organic Framework for Highly Selective Electroreduction of CO₂" makes me confused. Do the authors mean Oriented Electron Transportation?
2. In the "Investigation of the catalytic mechanism" part, diverse contrast samples like NNU-12, Co-TMCP and TMCP were prepared and tested. The structure images of these contrast samples would be better provided.
3. What is the main concern of choosing Zn- ϵ -Keggin cluster as the desired POM to construct various PMOFs? It should be discussed from the viewpoint of structure design or target properties.
4. Two kinds of additives (i.e. acetylene black (AB) and Nafion solution) were used to prepare the CCE working electrode. The authors should explain the reasons.
5. There are some unnecessary blank spaces like "Co- PMOF" in the sentence "All the M-PMOFs (i.e. Co- PMOF, Fe- PMOF, Ni- PMOF, and Zn- PMOF) samples".
6. In Supplementary Table 2, the compound name like Co-TCPOM listed in the table should be Co-PMOF, correct it.
7. Line 102: converse CO₂ to CO should be convert CO₂ to CO.

Reviewer #2 (Remarks to the Author):

This manuscript reports the synthesis and characterizations of a series of polyoxometalate-metalloporphyrin organic frameworks (M-PMOFs). The Co-PMOF is an efficient catalyst for the reduction of CO₂ into CO. This material represents a rare example of a POM-based material which can reduce CO₂. The originality is however lowered by the fact that the catalytic species is not the POM but the metalloporphyrin. Furthermore there are several problems which are listed below. Therefore I cannot recommend the publication of this manuscript in Nature Communications.

- 1) There is a total lack of characterizations of the materials. I could not even find the detailed formulae. The authors only use the abbreviation M-PMOFs throughout the manuscript. There are no elemental analysis, no interpretation of the IR spectra and of the TGA measurements.
- 2) Supplementary Figure 3 raises several problems. First it is really hard to see if the experimental PXRD patterns correspond to the simulated ones because of the very small sizes of the Figures. The Figures must be enlarged and shown on a single column and a zoom should be made for 2 Theta between 0 and 20 degrees. In the absence of elemental analysis, it is thus hard to conclude about the purity of the phase.
- 3) The stability of the M-PMOFs in a pH range from 5 to 11 and in the conditions of electrocatalysis cannot rely on a single experiment which is PXRD. Some peaks seem to be present at pH 5 for Co-PMOF and Ni-PMOF which are absent in the simulated powder pattern and could indicate the existence of an additional phase. The IR spectra of the solids must be recorded as well as an analysis of the material at least by EDX-chemical mapping and an analysis of the solution by ICP to check the absence of leaching.
- 4) TGA experiments are not sufficient and thermogravimetric analysis experiments should be performed in order to assess that the structures are stable up to 200°C.
- 5) For the comparison with Co-TMCP, can the authors precise if the compound is studied as a solid in the carbon ink or in solution? I suspect this compound to be soluble in 0.5 M KHCO₃. This must be clarified. A relevant comparison would be with a MOF containing Co-TMC such as MOF-525-Co.
- 6) The production of H₂ and CO (TON) as a function of time must be given.
- 7) An isotopic experiment using ¹³C CO₂ as substrate should be performed in order to prove the origin of CO.

Reviewer #3 (Remarks to the Author):

In this work, the authors reported several stable polyoxometalate-metalloporphyrin organic frameworks (i.e. Co-PMOF, Fe-PMOF, Ni-PMOF and Zn-PMOF) constructed from tetrakis[4-carboxyphenyl]-porphyrin-M (M-TCPP) linkers and reductive Zn- ϵ -Keggin clusters by hydrothermal method. Incorporation of reductive POM units and metalloporphyrins in the hybrid materials results in oriented electronic transportation channels, and thus the obtained PMOFs exhibited excellent electrochemical CO₂ reduction performances. In particular for Co-PMOF, it enables to convert selectively CO₂ to CO with remarkable faradaic efficiency of 99%, high TOF of 1656 h⁻¹ and excellent stability. The mechanisms of the catalysis are also clearly revealed by DFT calculations. Therefore, this work is very interesting and important. I therefore recommend to accept this manuscript for

publication in Nature Communications after minor revisions.

1. In this work, the samples were attached onto the surfaces of carbon clothes. The blank catalytic experiments with bare carbon cloth should be conducted, including some discussions.

2. In Supplementary Table 1, all of the listed materials are MOFs or COFs. The performances of other typical materials (e.g. carbon materials or metal nanoparticles, etc.) should also be compared.

3. Metalloporphyrinic MOFs are a class of promising materials for electrocatalysis. Some important works on metalloporphyrinic MOFs might be cited to support further the novelty of the PMOFs.

4. The detailed structures of NNU-12, Co-TMCP and TMCP should be discussed.

5. Minor errors should be corrected, such as the compound name of Co-TCPOM in Supplementary Table 2 should be Co-PMOF.

Responses to the Reviewers' Comments

Reviewer: 1

Comments to the Author:

Lan and coworkers have synthesized a series of novel polyoxometalate-metalloporphyrin organic frameworks (PMOFs) crystals and explored their applications in CO₂ reduction. By integration of Zn- ϵ -Keggin cluster and metalloporphyrin the electric conductivity and electron donating capability of the fabricated MOFs are substantially strengthened. Among these PMOFs, Co-PMOF exhibits remarkable faradaic efficiency up to 99% for CO and long-term stability. Further, the mechanisms of catalysis processes are also discussed and revealed by DFT calculations. The results reported here are quite interesting and inspiring. Therefore, I recommend that this work for publication after minor revision before the following questions are fully addressed in order to further improve the quality of the nice work.

1. One of the major problem to hinder the MOFs as electrocatalysts is their stability. The authors claimed that the PMOFs, for example, Co-PMOF exhibited excellent electrocatalytic stability (> 36 h) and provided solid evidences of the Co-PMOFs in solutions with various pH values, but before electrocatalytic reaction. I am very curious to know whether the authors have characterized Co-PMOF after electrocatalysis. Therefore, I would suggest the authors, at least, should provide the XRD pattern of Co-PMOF after 36 h.

Response:

Thanks for your kind suggestion. According to your suggestion, we have added the PXRD test of Co-PMOF after electrocatalysis. As shown in Fig. S27, the PXRD patterns of Co-PMOF agree well with the simulated one, indicating Co-PMOF can maintain the integrity of its structure after electrocatalysis. We have now included the discussion in the revised Supporting Information (Fig. S27, page 21, line 5).

Supplementary Figure 27. PXRD patterns of Co-PMOF after electrochemical experiment.

2. Concerning the mechanism proposed in Figure 4d, Co(I) species are involved during the electrocatalytic process. If the authors can provide direct evidences about the existence of the Co(I), such as EPR experiment, the proposed mechanism will be more solid.

Response:

Thanks for your suggestion. Co(I) is important in our proposed mechanism. CV curves and EPR tests are two promising characterization methods for the detection of Co(I) in experiment. As we mentioned in the first version of the manuscript, in Fig. S37, the cyclic voltammetry (CV) curves of Co-PMOF in N₂-saturated 0.5 M KHCO₃ show a broad cathodic wave centered at -0.45 V versus reversible hydrogen electrode (RHE). Compared with reported works, its position (centered at -0.45 V) falls within the general potential range for Co^{II}/Co^I observed for many Co-centered macrocycles, like cobalt porphyrins. For example, previous electrochemical studies of cobalt porphyrins reported by Morris's group have attributed a cathodic wave at approximately -0.5 V vs RHE to the reduction of Co(II) center to Co(I) (*J. Am. Chem. Soc.* **2014**, 136, 2464-2472) (**Figure R1-1**). Besides, Yang et al. discovered similar phenomenon for the H₄TCPP-Co at approximately -0.4 V vs RHE in the reduction of Co(II) center to Co(I) in CO₂RR experiment (*J. Am. Chem. Soc.* **2015**, 137, 14129–14135) (**Figure R1-2**). In addition, Li et al. reported CoPPc/CNT as the CO₂RR electrocatalyst with a similar broad cathodic wave for Co^{II}/Co^I (*Chem*, **2017**, 3, 652–664). As for EPR tests, another powerful protocol for valance test, it is difficult to simulate the actual electrochemical measurement system in the EPR test process, such as under the condition of electrification, which leads to the rare use of EPR to characterize the catalyst under electrocatalysis conditions in reported works. Generally, most of works about electrochemical CO₂RR applied CV to *in-situ* characterize the catalytic process. Thanks to the reviewer's kind suggestion, if possible, we will combine the electrochemical measurement system with EPR equipments in the future to explore the application of EPR in electrochemical CO₂RR.

Now we have properly cited relative references in the revised manuscript (ref 12, 29, 37 and 48, page 10, line 14).

Supplementary Figure 37. CV curve of Co-PMOF in N₂-saturated 0.5 M KHCO₃.

Figure R1-1. Cyclic voltammogram of CoPIZA/FTO in 0.1 M LiClO₄/DMF from -0.4 V to -1.15 V vs. ferrocyanide at various scan rates (10 mV/sec to 1000 mV/sec). (-0.5 V vs. RHE for Co^{II}/Co^I was calculated based on ferrocyanide) (Refer to *J. Am. Chem. Soc.* **2014**, 136, 2464-2472).

Figure R1-2. CV scans of the Co-TCPP linker drop cast on titanium foil at 100 mV/s under a CO₂ atmosphere also displayed a cathodic peak at -0.4 V vs. RHE preceding an irreversible catalytic peak (Refer to *J. Am. Chem. Soc.* **2015**, 137, 14129–14135).

3. The title “Oriented Electron Transmission in POM-Metalloporphyrin Organic Framework for Highly Selective Electroreduction of CO₂” makes me confused. Do the authors mean Oriented Electron Transportation?

Response:

Thanks for your suggestion. In this work, we applied polyoxometalates (POMs) and metalloporphyrins to construct polyoxometalate-metalloporphyrin organic frameworks. In the structure, reductive polyoxometalates mainly composed of low

valent metal ions, such as Zn- ϵ -Keggin cluster ($\{\epsilon\text{-PMo}_8^{\text{V}}\text{Mo}_4^{\text{VI}}\text{O}_{40}\text{Zn}_4\}$, including eight Mo^{V} atoms), are usually electron-rich aggregates and can easily offer electrons when triggered by redox reaction or bias stimulus. Co-porphyrin, where inherent macrocycle conjugated π -electron system is very beneficial for electron mobility and Co(II) enables to be reduced to Co(I) during the process in many references (as mentioned in Response 1). The connection of POM and metalloporphyrin will presumably create an oriented electron transportation pathway under the motivation of electric field, abundant electrons flowing from POM cluster to metalloporphyrin motif can guarantee and facilitate the fulfillment of multiple electron migration process of CO_2RR electrocatalysis.

Now we have added relative references and discussion in the revised Supporting Information (page 4, line 10).

4. In the “Investigation of the catalytic mechanism” part, diverse contrast samples like NNU-12, Co-TMCP and TMCP were prepared and tested. The structure images of these contrast samples would be better provided.

Response: Per suggestion, we have provided the structure images of NNU-12, Co-TMCP and TMCP and added relative structure descriptions in the revised Supporting Information (Fig. S28 and S29, page 22, line 1).

Supplementary Figure 28. The structure images of NNU-12. (a) Secondary building block. (b) Basic construction unit. (c) 3D framework. (d) Six-fold interpenetrated structure with a *dia* topology. As presented in the image, each BCPT^{2-} ligand connects two Zn- ϵ -Keggin segments and each Zn- ϵ -Keggin connects four ligands, which generates a 3D framework with six-fold interpenetrated structure with a *dia* topology. NNU-12 contains the same Zn- ϵ -Keggin unit as Co-PMOF while the ligand is different.

Supplementary Figure 29. The structure images of TMCP and Co-TMCP. (a) TMCP. (b) Co-TMCP. TMCP is a kind of ester compound and Co-TMCP is a kind of Co-centered macrocycle.

5. What is the main concern of choosing Zn- ϵ -Keggin cluster as the desired POM to construct various PMOFs? It should be discussed from the viewpoint of structure design or target properties.

Response:

Thanks for your insightful suggestion. The Zn- ϵ -Keggin cluster $\{\epsilon\text{-PMo}_8^{\text{V}}\text{Mo}_4^{\text{VI}}\text{O}_{40}\text{Zn}_4\}$ is embedded by four Zn^{2+} located in a regular tetrahedral arrangement, which offers a 4-connected mode to form interesting structures with outstanding stability. As a kind of reductive POMs composed of low valent metal ions (including eight Mo^{V} atoms), Zn- ϵ -Keggin cluster is electron-rich aggregate and can easily offer electrons when triggered by redox reaction or bias stimulus. The connection of Zn- ϵ -Keggin cluster and metalloporphyrin will presumably create an oriented electron transportation pathway under the motivation of electric field, abundant electrons flowing from Zn- ϵ -Keggin cluster to metalloporphyrin motif can guarantee and facilitate the fulfillment of multiple electron migration process of CO_2RR electrocatalysis. Hence in the structure design, we deduce that reductive polyoxometalate-metalloporphyrin organic frameworks will probably be promising candidates to enhance the efficiency and selectivity of CO_2RR . As a proof-of-concept, Co-PMOF exhibits remarkable faradaic efficiency ($\text{FE}_{\text{CO}} > 94\%$) over a wide potential range (-0.8 to -1.0 V) in this work. Its best FE_{CO} can reach up to 99% (highest in reported MOFs) and it exhibits a high TOF of 1656 h^{-1} and excellent catalysis stability ($> 36 \text{ h}$).

6. Two kinds of additives (i.e. acetylene black (AB) and Nafion solution) were used to prepare the CCE working electrode. The authors should explain the reasons.

Response:

Thanks for your kind suggestion. Given the poor intrinsic electrical conductivity of MOFs, acetylene black (AB) was introduced to mix with the as-synthesized MOFs to improve the conductivity. Nafion solution was introduced as a kind of MOF dispersion solution generally applied in many reported works, which can form a homogeneous ink with MOF and further help to attach onto the surface of carbon cloth.

Now we have added relative discussion in the revised manuscript (page 12, line 23).

7. There are some unnecessary blank spaces like “Co- PMOF” in the sentence “All the M-PMOFs (i.e. Co- PMOF, Fe- PMOF, Ni- PMOF, and Zn- PMOF) samples”.

Response: Per suggestion, we have deleted the unnecessary blank spaces and checked through the manuscript (page 3, line 17).

8. In Supplementary Table 2, the compound name like Co-TCPOM listed in the table should be Co-PMOF, correct it.

Response: Per suggestion, we have corrected the compound name “Co-TCPOM” to “Co-PMOF” and checked through the manuscript (Table S2, page 28).

9. Line 102: converse CO₂ to CO should be convert CO₂ to CO.

Response: Per suggestion, we have corrected the word “converse” to “convert” and checked through the manuscript (page 3, line 14).

Reviewer: 2

Comments to the Author:

This manuscript reports the synthesis and characterizations of a series of polyoxometalate-metalloporphyrin organic frameworks (M-PMOFs). The Co-PMOF is an efficient catalyst for the reduction of CO₂ into CO. This material represents a rare example of a POM-based material which can reduce CO₂. The originality is however lowered by the fact that the catalytic species is not the POM but the metalloporphyrin. Furthermore there are several problems which are listed below. Therefore I cannot recommend the publication of this manuscript in Nature Communications.

1. There is a total lack of characterizations of the materials. I could not even find the detailed formulae. The authors only use the abbreviation M-PMOFs throughout the manuscript. There are no elemental analysis, no interpretation of the IR spectra and of the TGA measurements.

Response:

Per suggestion, in this version, we have added detailed structure and stability characterizations (IR, TGA, EA and ICP) of M-PMOF and provided relative discussions in both revised manuscript and Supporting Information.

For the formulae, in the first version of the manuscript, we have provided the empirical formulae of Co-PMOF (C₄₈H₂₆CoMo₂₄N₄O₈₉P₂Zn₈), Fe-PMOF (C₄₈H₂₆FeMo₂₄N₄O₈₉P₂Zn₈), Ni-PMOF (C₄₈H₂₄Mo₂₄N₄NiO₈₈P₂Zn₈) and Zn-PMOF (C₁₂H_{6.5}Mo₆NO_{22.25}P_{0.5}Zn_{2.25}) in Supplementary Table 2. For the formulae of M-PMOF, all of these M-PMOF are iso-reticular in structure and only have minor difference in centered metal ions. Their structures have been well-defined in the SXRD tests. Based on the elemental analyses and SXRD tests, the calculated formulae was [PMo^V₈Mo^{VI}₄O₃₅(OH)₅Zn₄]₂[M-TCPP][2H₂O][1.5TBAOH] (M = Fe, Co, Ni and Zn).

The calculated formulae were further supported by TGA test (Fig. S8). Taking Fe-PMOF for example, in the test, about 6.8% mass loss at temperature range from 0 to 200 °C is attributed to the loss of guest molecules, which matches well with the content of guest molecules in Fe-PMOF.

Now we have added relative discussion in the revised manuscript (page 11, line 31) and Supporting Information (page 9, line 2).

For IR tests, relative IR peaks for Co-PMOF, Fe-PMOF, Ni-PMOF and Zn-PMOF are added in the syntheses part in the revised manuscript (page 11, line 24). The peak attribution in the IR spectra is discussed in Fig. S7 (Supporting Information, page 8, line 3).

The thermal stability of M-PMOF samples is studied by thermogravimetric analyses (TGA) in O₂ atmosphere. Under O₂ atmosphere, all these M-PMOF samples can be stable at temperatures higher than 200 °C. Taking Co-PMOF for example, about 4% mass loss at temperature range from 0 to 100 °C is attributed to the loss of guest molecules. After 200 °C, the framework of Co-PMOF starts to collapse and ends at about 500 °C (Fig. S8). To further certify the thermal stability, PXRD patterns of M-PMOF samples treated at 200 °C in the presence of ultrapure O₂ were tested. As

shown in Fig. S9, the PXRD patterns of M-PMOF samples still agree well with the simulated ones.

Now we have added relative discussion in the revised manuscript (page 4, line 20) and Supporting Information (page 10, line 3).

2. Supplementary Figure 3 raises several problems. First it is really hard to see if the experimental PXRD patterns correspond to the simulated ones because of the very small sizes of the Figures. The Figures must be enlarged and shown on a single column and a zoom should be made for 2 Theta between 0 and 20 degrees. In the absence of elemental analysis, it is thus hard to conclude about the purity of the phase.

Response: Thanks for your suggestion. We have enlarged the figures and a zoom have been made for 2 Theta between 3 and 20 degrees (catalyst has no peaks before 3 degrees) for all of the M-PMOF (Co-PMOF, Fe-PMOF, Ni-PMOF and Zn-PMOF) (Fig. S3-6). As showed in the image, all of the PXRD patterns of the as-synthesized samples match well with the simulated ones, which indicates the high purity and crystalline of these M-PMOF. Besides, the elemental analysis tests mentioned in Response 1 further prove the purity of the phase (Supporting Information, page 6-7).

Supplementary Figure 3. PXRD patterns of Co-PMOF. (a) 2 Theta ranges from 3° to 50°. (b) 2 Theta ranges from 3° to 20°. “Sim”: simulated pattern and “Exp”: as-synthesized sample.

Supplementary Figure 4. PXRD patterns of Fe-PMOF. (a) 2 Theta ranges from 3° to 50°. (b) 2 Theta ranges from 3° to 20°. “Sim”: simulated pattern and “Exp”: as-synthesized sample.

Supplementary Figure 5. PXRD patterns of Ni-PMOF. (a) 2 Theta ranges from 3° to 50°. (b) 2 Theta ranges from 3° to 20°. “Sim”: simulated pattern and “Exp”: as-synthesized sample.

Supplementary Figure 6. PXRD patterns of Zn-PMOF. (a) 2 Theta ranges from 3° to 50°. (b) 2 Theta ranges from 3° to 20°. “Sim”: simulated pattern and “Exp”: as-synthesized sample.

3. The stability of the M-PMOFs in a pH range from 5 to 11 and in the conditions of electrocatalysis cannot rely on a single experiment which is PXRD. Some peaks seem to be present at pH 5 for Co-PMOF and Ni-PMOF which are absent in the simulated powder pattern and could indicate the existence of an additional phase. The IR spectra of the solids must be recorded as well as an analysis of the material at least by EDX-chemical mapping and an analysis of the solution by ICP to check the absence of leaching.

Response:

Thanks for your kind suggestion. In the new version, stability tests of M-PMOF are carefully characterized with PXRD, IR, EDX-chemical mapping and ICP tests and the obtained data are well-organized and discussed.

For PXRD tests, the images are enlarged and showed on a single column (Fig. S10-13). In the enlarged images, the PXRD patterns of M-PMOF (i.e. Co-PMOF, Fe-PMOF, Ni-PMOF and Zn-PMOF) after treating in acid, base and 0.5 M KHCO₃ solutions for 24 h match well with the simulated ones, which verify the high chemical stability of M-PMOF (Supporting Information, pages 10-12).

As for the claim “Some peaks seem to be present at pH 5 for Co-PMOF and Ni-PMOF which are absent in the simulated powder pattern and could indicate the existence of an additional phase”, some peaks of M-PMOF for simulated patterns are weak in grouped figures in the first version and they can be visible in the enlarged figures in this one. After carefully comparison, the simulated PXRD pattern is

consistent well with patterns obtained at pH = 5 (Figs. S10-13). Besides, we have added the IR spectra studies of M-PMOF after stability test in acid, base and 0.5 M KHCO₃ solutions. As shown in Fig. S14, the IR spectra of M-PMOF in acid, base and 0.5 M KHCO₃ solutions have negligible change compared with the as-synthesized ones, indicating M-PMOF can maintain the integrity of their structures after chemical stability tests. We have added relative discussion in the revised Supporting Information (page 12, line 7).

Moreover, Fig. S15 presents SEM and energy dispersive X-ray (EDX) elemental mapping images for Co-PMOF, Fe-PMOF, Ni-PMOF and Zn-PMOF after stability test in acid solution. In SEM tests, M-PMOF are all in regular cubic shapes and EDX tests show that metal elements distribute uniformly on the M-PMOF crystals after stability tests. Further proved by the ICP leaching test, negligible leaching metal ions were detected in the solution after chemical stability tests.

Now we have added relative discussion in the revised Supporting Information (page 13, line 4).

Supplementary Figure 10. PXRd patterns of Co-PMOF in acid, base and 0.5 M KHCO₃ solutions. (a) 2 Theta ranges from 3° to 50°. (b) 2 Theta ranges from 3° to 20°. “Sim”: simulated pattern and “Exp”: as-synthesized sample.

Supplementary Figure 11. PXRd patterns of Fe-PMOF in acid, base and 0.5 M KHCO₃ solutions. (a) 2 Theta ranges from 3° to 50°. (b) 2 Theta ranges from 3° to 20°. “Sim”: simulated pattern and “Exp”: as-synthesized sample.

Supplementary Figure 12. PXRD patterns of Ni-PMOF in acid, base and 0.5 M KHCO_3 solutions. (a) 2 Theta ranges from 3° to 50° . (b) 2 Theta ranges from 3° to 20° . “Sim”: simulated pattern and “Exp”: as-synthesized sample.

Supplementary Figure 13. PXRD patterns of Zn-PMOF in acid, base and 0.5 M KHCO_3 solutions. (a) 2 Theta ranges from 3° to 50° . (b) 2 Theta ranges from 3° to 20° . “Sim”: simulated pattern and “Exp”: as-synthesized sample.

Supplementary Figure 14. IR spectra of M-PMOF in acid, base and 0.5 M KHCO_3 solutions at room temperature. (a) Co-PMOF. (b) Fe-PMOF. (c) Ni-PMOF. (d) Zn-PMOF.

Supplementary Figure 15 SEM image and EDX elemental mapping of M-PMOF after stability tests (in acid at room temperature). (a) Co-PMOF. (b) Fe-PMOF. (c) Ni-PMOF. (d) Zn-PMOF.

4. TGA experiments are not sufficient and thermogravimetric analysis experiments should be performed in order to assess that the structures are stable up to 200 °C.

Response: Thanks for your suggestion. We have added the thermogravimetric analysis studies of M-PMOF. M-PMOF samples were placed in a tubular furnace and carbonized in the presence of ultrapure O₂ at 200 °C with a heating rate of 5 °C min⁻¹. As shown in Fig. S9, the PXRD patterns of M-PMOF still agree well with the simulated ones, indicating the structures of M-PMOF enable to be stable up to 200 °C (Fig. S9).

Now we have added relative discussion in the revised manuscript (page 4, line 20) and Supporting Information (page 10, line 3).

Supplementary Figure 9. PXRD patterns of M-PMOF after thermal treatment. (a) Co-PMOF. (b) Fe-PMOF. (c) Ni-PMOF. (d) Zn-PMOF. All the samples were treated at 200 °C in O₂ atmosphere with a heating rate of 5 °C min⁻¹. “Sim”: simulated pattern and “Exp”: as-synthesized sample.

5. For the comparison with Co-TMCP, can the authors precise if the compound is studied as a solid in the carbon ink or in solution? I suspect this compound to be soluble in 0.5 M KHCO₃. This must be clarified. A relevant comparison would be with a MOF containing Co-TMC such as MOF-525-Co.

Response: Thanks for your insightful suggestion. Co-TMCP is a kind of ester compound, which is insoluble in water or 0.5 M KHCO₃. Co-TMCP is fuchsia. If it is dissolved in the solution, the color might have some change. After electrocatalysis test, the solution has no color change (Fig. S38). Recycle test was further preformed to verify the insolubility of Co-TMCP and an efficiency of 43.4% at -0.8 V vs. RHE similar to the original one (43.6% at -0.8 V vs. RHE) was achieved (Supporting Information, page 27, line 5).

Moreover, MOF-525(Co) was prepared as relevant comparison to further support the catalytic mechanism of M-PMOF. MOF-525(Co) has similar ligand as Co-PMOF and is constructed with Zr₆O₄(OH)₄ unit without POM. In the structure, each TCPP⁴⁻ ligand connects with four Zr₆O₄(OH)₄ units to generate a 3D framework with a *ftw* topology. The comparison between MOF-525(Co) and Co-PMOF can further reveal the function of POM in the electrochemical CO₂RR. MOF-525(Co) is successfully prepared and certified by the PXRD tests (Fig. S32). The electrochemical CO₂RR of MOF-525(Co) is measured adopting the same testing methods. As shown in Fig. S33, MOF-525(Co) shows lower FE_{CO} (47.9% at -0.8 V) than Co-PMOF (98.7% at -0.8 V), which might be attributed to the poorer proton and electron transfer efficiency. This indicates POMs actually act as electron-rich aggregates in the catalytic mechanism of M-PMOF. In Co-PMOF, the connection of POM and metalloporphyrin will create an

oriented electron transportation pathway under the motivation of electric field, abundant electrons flowing from POM cluster to metalloporphyrin motif can guarantee and facilitate the fulfillment of multiple electron migration process of CO₂RR to achieve higher performance than MOF-525(Co).

Now we have added relative discussion in the revised manuscript (page 8, line 30) and Supporting Information (page 24, line 3).

Supplementary Figure 38. The photo images of Co-TMCP before and after CO₂RR test. (a) Co-TMCP powder. (b) Co-TMCP loaded carbon cloth electrode before CO₂RR test. (c) Co-TMCP loaded carbon cloth electrode after CO₂RR test. (d) The electrolyte before CO₂RR test. (e) The electrolyte after CO₂RR test.

Supplementary Figure 31. (a) The coordination environment of Zr₆O₄(OH)₄ unit in MOF-525(Co). (b) MOF-525(Co). (c) The coordination environment of Zn-ε-Keggin unit in Co-PMOF. (d) Co-PMOF. Co-PMOF was constructed by the 4-connected TCPP linkers and zigzag POM chains. In the structure of MOF-525(Co), each Zr₆O₄(OH)₄ unit connects with twelve TCPP⁴⁻ ligand to generate a 3D framework with a *ftw* topology.

Supplementary Figure 32. PXRD patterns of MOF-525(Co). “Sim”: simulated pattern and “Exp”: as-synthesized sample.

Supplementary Figure 33. Faradaic efficiencies of MOF-525(Co) and Co-PMOF. (a) Faradaic efficiencies (CO and H₂) of MOF-525(Co). (b) Faradaic efficiencies (CO) of MOF-525(Co) and Co-PMOF at different applied potentials in CO₂-saturated 0.5 M KHCO₃ aqueous solution.

6. The production of H₂ and CO (TON) as a function of time must be given.

Response: Thanks for your suggestion. According to your suggestion, we have added the calculation in the Fig. S24 and provided a discussion in the revised manuscript (page 6, line 32) and supporting information (page 2, line 29; page 19, line 3).

Turnover number (TON) is defined as the mole of reduction product generated per electrocatalytic active site over a given period of time.

The TON for CO was calculated as follows:

$$\text{TON} = \frac{Q \times FE_{\text{CO}}(\text{average})}{2F \times n_{\text{tot}}}$$

Supplementary Figure 24. Plots of CO and H₂ evolving turnover number versus time for Co-PMOF. As shown in the images, the TON (H₂) is close to 0 owing to its low efficiency for H₂ generation. Notably, the TON (CO) is as high as 7693 in just 5 h and can reach up to 53433 after 35 h.

7. An isotopic experiment using ¹³CO₂ as substrate should be performed in order to prove the origin of CO.

Response: Thanks for your insightful suggestion. According to your suggestion, an isotopic experiment that using ¹³CO₂ as substrate was performed under identical reaction conditions to reveal the carbon source of CO. The products were analyzed by gas chromatography and mass spectra. As shown in Fig. S22, the peak at m/z = 29 is assigned to ¹³CO, indicating that the carbon source of CO indeed derives from the CO₂ used.

Now we have added the data and relative discussion in the revised manuscript (page 6, line 23) and Supporting Information (page 18, line 1).

Supplementary Figure 22. The mass spectra of ¹³CO recorded under ¹³CO₂ atmosphere. The peak at m/z = 29 is assigned to ¹³CO.

Reviewer: 3

Comments to the Author:

In this work, the authors reported several stable polyoxometalate-metalloporphyrin organic frameworks (i.e. Co-PMOF, Fe-PMOF, Ni-PMOF and Zn-PMOF) constructed from tetrakis[4-carboxyphenyl]-porphyrin-M (M-TCPP) linkers and reductive Zn- ϵ -Keggin clusters by hydrothermal method. Incorporation of reductive POM units and metalloporphyrins in the hybrid materials results in oriented electronic transportation channels, and thus the obtained PMOFs exhibited excellent electrochemical CO₂ reduction performances. In particular for Co-PMOF, it enables to converse selectively CO₂ to CO with remarkable faradaic efficiency of 99%, high TOF of 1656 h⁻¹ and excellent stability. The mechanisms of the catalysis are also clearly revealed by DFT calculations. Therefore, this work is very interesting and important. I therefore recommend to accept this manuscript for publication in Nature Communications after minor revisions.

1. In this work, the samples were attached onto the surfaces of carbon clothes. The blank catalytic experiments with bare carbon cloth should be conducted, including some discussions.

Response: Per suggestion, we conducted catalytic experiments to reveal the CO₂RR activity of the bare carbon cloth. As shown in Fig. S20, the pure carbon cloth has no CO₂RR activity compared with Co-PMOF, Fe-PMOF, Ni-PMOF and Zn-PMOF. Therefore, the application of carbon cloth as the substrate has no effect on the CO₂RR properties of M-PMOF.

Now we have added relative discussion in the revised Supporting Information. (page 17, line 3).

Supplementary Figure 20. Faradaic efficiencies (CO and H₂) of pure carbon cloth at different applied potentials in CO₂-saturated 0.5 M KHCO₃ aqueous solution.

2. In Supplementary Table 1, all of the listed materials are MOFs or COFs. The performances of other typical materials (e.g. carbon materials or metal nanoparticles, etc.) should also be compared.

Response: Thanks for your suggestion. We have listed other typical materials (e.g., Ni SAs/N-C, nanoporous Ag, Pd nanoparticles, N-based silver catalyst and CATPyr/CNT, etc.) in Supplementary Table 1 and added a discussion in the revised Supporting Information (Table S1, page 16, line 1).

3. Metalloporphyrinic MOFs are a class of promising materials for electrocatalysis. Some important works on metalloporphyrinic MOFs might be cited to support further the novelty of the PMOFs.

Response: Per suggestion, we have carefully searched metalloporphyrinic MOFs based references and some important works are properly cited in the revised manuscript (page 2, line 39).

4. The detailed structures of NNU-12, Co-TMCP and TMCP should be discussed.

Response: Per suggestion, we have provided the structure images of NNU-12, Co-TMCP and TMCP and added relative structure descriptions in the revised Supporting Information (Fig. S28 and S29) (page 22, line 1).

Supplementary Figure 28. The structure images of NNU-12. (a) Secondary building block. (b) Basic construction unit. (c) 3D framework. (d) Six-fold interpenetrated structure with a *dia* topology. As presented in the image, each BCPT^{2-} ligand connects two Zn- ϵ -Keggin segments and each Zn- ϵ -Keggin connects four ligands, which generates a 3D framework with six-fold interpenetrated structure with a *dia* topology. NNU-12 contains the same Zn- ϵ -Keggin unit as Co-PMOF while the ligand is different.

Supplementary Figure 29. The structure images of TMCP and Co-TMCP. (a) TMCP. (b) Co-TMCP. TMCP is a kind of ester compound and Co-TMCP is a kind of Co-centered macrocycle.

5. Minor errors should be corrected, such as the compound name of Co-TCPOM in Supplementary Table 2 should be Co-PMOF.

Response: Per suggestion, we have corrected the compound name “Co-TCPOM” to “Co-PMOF” and checked through the manuscript (Table S2, page 28).

REVIEWERS' COMMENTS:

Reviewer #1 (Remarks to the Author):

After looking through the reply, the authors have adequately answered all the concerns raised by the reviewers. The quality of revised version has been substantially improved. The presented results of PMOFs for electrocatalytic reduction of CO₂ are important, which will inspire the MOF community to develop new type of MOFs for electrocatalysis. I therefore recommend the revised version for the publication on Nature Communications without further change.

Reviewer #2 (Remarks to the Author):

I am fully satisfied by the answers and the additional experiments provided by the authors and now recommend the publication of this paper with no further corrections.

Reviewer #3 (Remarks to the Author):

I suggest to accept this manuscript for publication.

Responses to the Reviewers' Comments

Reviewer: 1 (Remarks to the Author):

After looking through the reply, the authors have adequately answered all the concerns raised by the reviewers. The quality of revised version has been substantially improved. The presented results of PMOFs for electrocatalytic reduction of CO₂ are important, which will inspire the MOF community to develop new type of MOFs for electrocatalysis. I therefore recommend the revised version for the publication on Nature Communications without further change.

Reviewer: 2 (Remarks to the Author):

I am fully satisfied by the answers and the additional experiments provided by the authors and now recommend the publication of this paper with no further corrections.

Reviewer: 3 (Remarks to the Author):

I suggest to accept this manuscript for publication.

We wanted to thank you and the referees for the positive comments for our works.